# *Lactobacillus gasseri* RW2014 Ameliorates Hyperlipidemia by Modulating Bile Acid Metabolism and Gut Microbiota Composition in Rats

**DOI:** 10.3390/nu14234945

**Published:** 2022-11-22

**Authors:** Xianping Li, Yuchun Xiao, Yuanming Huang, Liqiong Song, Mengde Li, Zhihong Ren

**Affiliations:** 1State Key Laboratory of Infectious Disease Prevention and Control, National Institute for Communicable Disease Control and Prevention, Chinese Center for Disease Control and Prevention, Beijing 102206, China; 2National Engineering Center of Dairy for Early Life Health, Beijing Sanyuan Foods Co., Ltd., Beijing 100163, China; 3School of Computer Science and Information Engineering, Hefei University of Technology, Hefei 230002, China

**Keywords:** *Lactobacillus gasseri*, probiotics, hyperlipidemia, bile acids, metabolism, gut microbiota

## Abstract

Hyperlipidemia is a leading risk of cardiovascular and cerebrovascular disease. Dietary supplementation with probiotics has been suggested as an alternative intervention to lower cholesterol. In the current study, we isolated a strain of *Lactobacillus gasseri* RW2014 (LGA) from the feces of a healthy infant fed with breast milk, and it displayed bile salt hydrolase (BSH) activity. Using this strain we determined its cholesterol-lowering and fatty liver-improving functions. SD rats were randomly divided into four groups. The control rats were fed a commercial chow diet and the other three groups were fed a high-fat diet (HFD) for a 7-week experiment period. After two weeks of feeding, the rats in PBS, simvastin, and LGA group were daily administered through oral gavage with 2 mL PBS, simvastin (1 mg/mL), and 2 × 10^9^ CFU/mouse live LGA in PBS, respectively. After five weeks of such treatment, the rats were euthanized and tissue samples were collected. Blood lipid and inflammatory factors were measured by ELISA, gut microbiota was determined by 16S rRNA sequencing, and bile acids profiles were detected by metabolomics. We found that LGA group had lower levels of blood cholesterol and liver steatosis compared to the simvastin group. LGA also significantly reducedthe levels of inflammatory factors in the serum, including TNFα, IL-1β, MCP-1, IL-6, and exotoxin (ET), and increased the levels of short-chain fatty acids in feces, including isobutyric acid, butyric acid, isovaleric acid, valeric acid, and hexanoic acid. In addition, LGA altered the compositions of gut microbiota as manifested by the increased ratio of Firmicutes/Bacteroides and the relative abundance of *Blautia* genus. Targeted metabolomics results showed that bile acids, especially free bile acids and secondary bile acids in feces, were increased in LGA rats compared with the control rats. Accordingly, the rats administrated with LGA also had a higher abundance of serum bile acids, including 23-norcholic acid, 7-ketolithocholic acid, β-muricholic acid, cholic acid, and deoxycholic acid. Together, this study suggests that LGA may exert a cholesterol-lowering effect by modulating the metabolism of bile acids and the composition of gut microbiota.

## 1. Introduction

Hyperlipidemia is a leading risk of cardiovascular and cerebrovascular disease, of which myocardial infarction and cerebral thrombosis are the leading causes of death and disability. Thus, the wide spread of hyperlipidemia and hepatic steatosis (fatty liver) has become a serious concern in public health [1]. Reducing cholesterol in the early phases of cardiovascular and cerebrovascular diseases has been shown to be effective in preventing disease development. The use of hypolipidemic drugs, such as simvastin, has been a primary treatment for majority of patients with high blood lipids [2]. However, some people who have slight hyperlipidemia and a mild to moderate fatty liver are psychologically reluctant to take the drugs. Moreover, side effects due to the long-term use of statins are well known. Therefore, efforts have been made to identify effective alternative remedies for reducing blood lipids with minimum side effects. Preliminary studies suggest that probiotics might be one such candidate worth looking into.

Probiotics are defined as “live microorganisms that, when administered in adequate amounts, confer a health benefit on the host” [3] and have been widely used to improve metabolic diseases. Jeffrey I Gordon et al. demonstrated that obese patients had a different composition of gut microbiota compared to lean individuals, and the obesity symptoms were improved by the fecal microbiota transplantation from lean individuals [4]. Studies have shown that many probiotics have cholesterol-lowering properties and most of them are genera of *Lactobacilli* [5,6] and *Bifidobacterium* [7,8]. For example, *Lactobacillus paracasei*, *Lactobacillus rhamnosus,* and *Bifidobacterium animalis* were able to attenuate obesity comorbidities in high-fat diet-fed mice [9], *Bifidobacterium adolescentis* [10], *Lactobacillus crustorum* [11], *Lactobacillus curvatus*, and *Lactobacillus plantarum* [12] exerted anti-obesity effects by modulating the composition of gut microbiota. We previously reported that a strain of *Lactobacillus plantarum* HT121 isolated from the feces of wild Tibetan marmots without using antibiotics reduced serum cholesterol by altering the composition of gut microbiota and liver transcriptome [5]. The early study reported the hypocholesterolemic effect of *L. gasseri* SBT0270 in rats [13]; however, the exact mechanism for the cholesterol-lowering by *L. gasseri* remains unclear. Along these lines, in this study, we determined the probiotic character of *Lactobacillus gasseri* RW2014 (LGA), a strain isolated from the feces of a healthy breastfeeding infant. The results of the in vitro experiments suggest that LGA may have acid- and bile-tolerancing and cholesterol-lowering properties. LGA significantly reduced cholesterol, altered the composition of gut microbiota, and promoted the discharge of total bile acids as well as free bile acids in hypocholesterolemic rats. These results support *L. gasseri*, being a probiotic with strong potential for helping control hyperlipidemia.

## 2. Materials and Methods

### 2.1. Ethics Statement

The present study was approved by the Ethics Review Committee of the National Institute for Communicable Disease Control and Prevention at the Chinese Center for Disease Control and Prevention (Beijing, China).

### 2.2. Screening and BSH Detection of LGA

The LGA strain used in this study was isolated from the feces of a healthy, breast-feeding infant and was screened by an acid- and bile salt-tolerance and cholesterol assimilation assay, as described in our previous study [5]. Briefly, 100 μL of feces suspension was diluted in a serial 10-fold increment way and then coated over Man–Rogosa–Sharpe (MRS) plates and cultured in a carbon dioxide (CO2) incubator at 37 °C for 2 d. The tolerance assay against acid and bile salt was performed on an MRS plate containing 0.3% bovine bile salt at pH 2.0. The strains with good tolerance were screened and evaluated for cholesterol assimilation by WAKO LabAssay Cholesterol assay kit following the kit’s instructions. LGA strain was selected according to the results of acid- and bile salt-tolerance and cholesterol assimilation assay. The BSH activity detection was conducted as the following: a sterile round filter paper was placed on the surface of the MRS plate containing 0.3% Taurodeoxycholic acid (TDCA), 0.2% sodium thioglycolate, and 0.37 g/L Cacl_2_. An amount of 10 μL of suspension containing 10^7^ CFU live LGA was dripped to the filter paper, then the MRS plate was cultured at 37 °C for 2 d [14].

### 2.3. Hypercholesterolemic Rat Model and LGA Administration

SPF Sprague Dawley rats (220–240 g) were purchased from the Beijing Vital River Laboratory Animal Technology Co., Ltd. (Beijing, China). The rats were housed in the Animal Center of China CDC at a temperature of 24–26 °C with humidity of 40–60% under a 12 h light–dark cycle and had free access to water and food. After 7 days of acclamation, the rats were randomly divided into four groups. The rats in the control and PBS groups (*n* = 10 for each group) were fed a normal chow diet or high-fat diet (HFD), respectively, for 7 weeks. The rats in the simvastin group (*n* = 6) and LGA group (*n* = 10) were fed HFD for 7 weeks. After two weeks, the rats were administered the following daily by oral gavage for a further 5 weeks: 2 mL PBS in the control and PBS groups, 2 mL PBS containing simvastin (1 mg/mL) in the simvastin group, or 2 × 10^9^ CFU/mouse live LGA in the LGA group. Diet and water consumption and body weight were recorded twice a week.

### 2.4. Measurement of Serum Lipids and Inflammatory Factors

At the end of the study, rats were anesthetized with an intraperitoneal injection of 1% pentobarbital and then blood was collected by cardiac puncture. The serums were isolated for the analysis of lipid profile, including TG, CH, HDL-C, and LDL-C, and inflammatory factors, including TNFα, IL-6, MCP-1, IL-1β, and ET by ELISA kits (Beijing Luyuan Bode Biotechnology Co., Beijing, China), according to manufacturer’s instruction.

### 2.5. Histopathological Analysis

Liver tissues were collected and fixed in 10% formalin solution for 24 h. Fixed tissues were embedded in paraffin. Then the sliced sections of 4 μm were stained with Hematoxylin and Eosin solution (Wuhan Servicebio Technology Co., Ltd., Wuhan, China). The images of HE sections were acquired with a microscope and analyzed by CaseViewer software.

### 2.6. Measurement of Fecal SCFAs

Freshly collected feces were flash-frozen in liquid nitrogen and stored at −80 °C. Fecal SCFAs were determined by the Beijing BioTech Pack Technology Company Ltd., Beijing, China. In brief, 200 mg fecal sample was suspended in 500 μL water with pH 1.0, followed by adding 500 μL ethyl acetate, and then vortexed for 30 min. After centrifugation at 1200 rpm for 10 min, the concentrations of SCFAs in supernatants were detected by GC–MS. The analysis conditions were as follows: heated to the initial temperature at 50 °C for 1 min, increased to 170 °C at 12 °C/min, and then to 230 °C at 20 °C/min, which lasted for 3 min. The inlet temperature of 250 °C, the split ratio of 1/50, the flow rate of 1.2 mL/min, and the detector temperature of 230 °C were set.

### 2.7. Quantification Analysis of Bile Acids

The bile acid profile in fecal samples was analyzed by Novo Gene Genetics (Beijing, China) using the condition as prescribed previously [15]. In brief, 100 mg of feces sample was suspended in 500 μL precipitant and centrifugated at 12,000 rpm for 10 min. The supernatant was freeze-dried with a vacuum, and 100 μL 20% acetonitrile was used to dissolve the resulting sediment. The profile of bile acids in the solution was quantified using liquid chromatography (LC, UltiMate 3000 UHPLC, Thermo Scientifc^TM^, Waltham, MA, USA) and mass spectrometer (MS, Q ExactiveTM, Thermo Scientifc^TM^, USA). The bile acid concentrations were calculated based on the peak areas of the bile acids and external standards. The data were analyzed using the SIMCA (v. 14.1).

### 2.8. 16S rRNA Sequencing and Data Analysis

Fresh fecal samples collected as described above were used for 16S rRNA sequencing and the data analysis was performed by the Beijing Genomics Institute Co., Ltd. (Beijing, China). Briefly, V3 and V4 hypervariable regions of 16S rRNA were amplified by primer pairs: forward primer (5′-CCTACGGGNGGCWGCAG-3′) and reverse primer (5′-GACTACHVGGGTATCTAATCC-3′), and the amplicons and quality control of the raw data were performed on the Illumina Hiseq 2500 platform. The consensus sequence was generated by FLASH. The high-quality paired-end reads were combined into tags based on overlaps, which were clustered into an Operational Taxonomic Unit (OUT) using scripts of software USEARCH with a 97% threshold. The α diversity and principal coordinates analysis (PCoA) both were calculated and visualized by R software. Linear discriminant analysis effect size (LEfSe) analysis (http://huttenhower.sph.harvard.edu/galaxy/, accessed on 1 March 2021) was conducted online and nonparametric factorial Kruskal–Wallis and Wilcoxon rank-sum tests were used. The threshold of 2.0 for discriminative features was set. The original data were uploaded into the NCBI SRA database (PRJNA863678).

### 2.9. Metabolomics Analysis of Serum

Serum samples stored at −80 °C were used for metabolomics analysis by Beijing Genomics Institute Co., Ltd. (Beijing, China). Briefly, the serum was dried in an Eppendorf tube and then re-suspended. Chromatographic separation was conducted using an Acquity UPLC HSS T3 column (Waters Corp., Milford, CO, USA) in a Thermo Scientific TM Q Exactive TM system. The column elution condition of flow rate was 0.30 mL/min at 40 °C. The eluent was detected by the mass spectrometer. The capillary voltage was 3.0 KV and 2.5 KV in the positive model and negative model, respectively. Dry nitrogen was used as desolvation gas, with a flow rate of 800 L/h at 450 °C. The scan time was 0.15 s and inter scan delay was 0.02 s. Spectral data from 100 *m*/*z* to 1500 *m*/*z* were collected in centroid mode. The identification of putative metabolite was conducted using the Tandem MS databases and ChemSpider chemical structure database. Some metabolites of particular interest were verified in METLIN^®^ database using the observed *m*/*z* with 3 ppm of mass error constraint. To further analysis, the mzXML open data format was converted from raw data using ProteoWizard^®^. SIMCA-P was used to perform multivariate statistical analysis.

### 2.10. Statistical Analysis

Statistical analysis was conducted using GraphPad Prism (v.5.0). One-way analysis of variance (ANOVA) followed by Tukey’s test was used to compare the difference in lipid profiles, SCFAs, and inflammatory factors between groups. *p* < 0.05 was considered statistically significant.

## 3. Results

### 3.1. L. gasseri RW2014 Had the Character of Probiotics and Activity of Bile Salt Hydrolase (BSH)

Since probiotic BSH can hydrolyze conjugated BAs to release unconjugated BAs, which are more difficult to be reabsorbed and easier to be excreted in feces, probiotic BSH may contribute to reducing serum cholesterol. We examined the BSH activity of LGA in the in vitro tests and found that LGA was most efficient in acid-, bile-tolerance, and cholesterol-reducing property among all the candidate strains being screened (data not shown). Genome analysis indicated that LGA contained three genes present in BSH (Appendix A), the BSH activity was confirmed in the solid medium as the LGA-inoculation sites showed large hydrolytic rings, which were not found at the control spots (Appendix A).

### 3.2. Oral Administration of L. gasseri RW2014 Significantly Improved the Hyperlipidemia in Rat Model

To evaluate the effect of LGA on hyperlipidemia, rats fed a high-fat diet were orally administrated with LGA or PBS for 6 weeks. We found that LGA significantly decreased the serum levels of total cholesterol (CH), triglycerides (TG), and low-density lipoprotein (LDL-C) and significantly increased the high-density lipoprotein (HDL-C), compared with those hyperlipidemia rats treated with PBS (Figure 1A–D). The effect of LGA is equivalent to simvastin, the most commonly used drug for the treatment of hyperlipidemia. There was no difference in body weight among all four groups (Appendix A). HFD feeding increased liver weight, which tended to be reduced by LGA (Appendix A). To minimize the influence of intra group difference in body weight, the liver index (the ratio of liver weight to body weight) was calculated. The results showed that LGA administration significantly lowered the liver index to a level comparable to that of the normal control rats (Appendix A), while simvastin had no such effect on the liver index. These results indicate that LGA can improve both hyperlipidemia and fatty liver. The effect of LGA on the liver was confirmed by histology results. There were plenty of round bubbles in the HE sections of the PBS group, which was caused by lipid droplets being resolved by organic solvents during HE staining (Figure 1F). The rats in both the simvastin group (Figure 1G) and LGA group (Figure 1H) had significantly fewer round bubbles compared to the PBS group, while in contrast, there was no round bubble in the normal control group (Figure 1E).

### 3.3. Oral Administration of L. gasseri RW2014 Significantly Reduced the Levels of Serum Inflammatory Factors

Since high serum cholesterol is usually accompanied by increased inflammatory factors, we quantified the levels of IL-6, IL-1β, TNFα, MCP-1, and ET in serum by ELISA. The results showed that LGA significantly reduced the elevated concentrations of IL-6, IL-1β, TNFα, and MCP-1 induced by HCD. There were no differences in these inflammatory factors between simvastin and LGA groups (Figure 2A–D) except that the level of ET was lower in the LGA group than that in the simvastin group (Figure 2E).

### 3.4. Oral Administration of L. gasseri Restored the Levels of SCFAs

SCFAs are the metabolic products of prebiotics interacting with microbiota, and they have been shown to play an important role in reducing cholesterol. SCFAs can activate peroxisome proliferator-activated receptors (PPARs), such as ligands. Measuring the levels of SCFAs in fecal samples by gas chromatography–mass spectrometry, we found that LGA markedly elevated the levels of isovaleric acid, isobutyric acid, valeric acid, butyric acid, and hexanoic acid compared to PBS treatment (Figure 3C,D,F,G). However, neither HCD nor LGA affected the concentrations of propionic acid, acetic acid, and isovaleric acid (Figure 3A,B,E).

### 3.5. Oral Administration of L. gasseri RW2014 Altered Composition of Gut Microbiota

HCD diet changed the overall composition of intestinal microbiota (Figure 4A), i.e., increased the relative abundance of Firmicutes, decreased Bacteroides, and increased ratio of F/B (Firmicutes/Bacteroides) (Figure 4B). The five-week administration of LGA decreased the relative abundance of Firmicutes phylum and genus of *Lachnospiracea_incertae_sedis* and increased the Bacteroidetes phylum and genera of *Bacteroides* and *Parabacteroides* (Figure 4B). It is worth noting that HFD rats had a slightly higher relative abundance of *Blautia*, while LGA rats had a significantly lower level of *Blautia* (Appendix A). Interestingly, there was no significant difference in the relative abundance of *Lactobacillus* genus among the three groups (Appendix A).

### 3.6. Oral Administration of L. gasseri RW2014 Increased the Concentrations of Fecal Bile Acids

A total of 31 fecal bile acids were measured using gas chromatography–mass spectrometry. A heat map, including 14 conjugated bile acids and 17 free bile acids, is shown in Figure 5. HFD group had higher concentrations of conjugated bile acid in feces and LGA significantly decreased the concentration of conjugated bile acid compared with the PBS group (Figure 6A). A similar trend was also observed in the concentrations of primary bile acids (Figure 6D). However, free bile acids (Figure 6B) and secondary bile acids (Figure 6E) were markedly elevated in the LGA group compared with the PBS and control groups. Consequently, LGA treatment significantly increased the ratio of free/conjugated bile acids compared with the control or PBS group (Figure 6C). HFD feeding significantly decreased the ratio of secondary/primary bile acids, while LGA administration effectively prevented this change (Figure 6F). HFD feeding did not alter the concentrations of lithocholic acid, isolithocholic acid, and allolithochoic acid, but LGA significantly elevated the concentrations of three bile acids in feces (Figure 6G–I). The concentrations of total bile acids in feces in each group were calculated and the rats in the LGA group and control group had the highest and lowest levels of total bile acids in feces, respectively (Figure 6J), indicating that HFD feeding caused more bile acids synthesized and excreted through feces and LGA supplement promoted more free and secondary bile acids excreted into feces.

### 3.7. The Effect of Oral Administration of L. gasseri RW2014 on Serum Metabolism

Three independent groups can be seen in PLS-DA score plot (Figure 7A). A total of 119 metabolites were identified (Appendix A). The results showed that the levels of 23−norcholic acid, 7−ketolithocholic acid, beta−muricholic acid, cholic acid, deoxycholic acid, and taurocholic acid sodium salt hydrate in LGA rats were significantly higher than those in PBS or control rats, and HFD feeding did not affect the metabolism of the bile acids above mentioned (Figure 7B–F,J). LGA also prevented HCD-induced elevation in the levels of glycoursodeoxycholic acid and taurochenodeoxycholic acid (Figure 7G,I). Both HCD and LGA significantly lowered the concentrations of tauro−alpha−muricholic acid sodium salt in the serum (Figure 7H).

## 4. Discussion

In this study, we demonstrated that LGA supplementation not only significantly reduced the levels of TG, CH, and LDL, but it also enhanced the HDL levels in hyperlipidemia rats, which is worth noting since most probiotics did not affect HDL [6]. Moreover, LGA was similarly effective in improving hyperlipidemia compared to simvastin, indicating that LGA may have the potential of being developed into a cholesterol-lowering drug.

Inflammatory factors were proven to be closely related to obesity. Obese patients often have hyperlipidemia. Hyperlipidemia is a major risk factor for atherosclerosis, and the levels of pro-inflammatory cytokines, such as TNF-α, IL-1β, and IL-6, are usually increased and play a vital role during the early progression of atherosclerotic plaques [16,17]. In this study, the hyperlipidemia rats had higher levels of inflammatory factors, which were reduced by both LGA and simvastin administration, indicating that LGA supplementation may decrease the risk of atherosclerosis in the hyperlipidemia rat. Endotoxin (i.e., lipopolysaccharide), a major outer membrane component of Gram-negative bacteria, can translocate through damaging gut barrier into the blood circulation and activate Toll-like receptor 4 signaling, further causing chronic inflammation and related diseases, including atherosclerotic cardiovascular diseases, diabetes, obesity, and metabolic syndrome [18,19]. Therefore, healthy individuals usually have a low level of circulating lipopolysaccharide activity. In this study, both LGA and simvastin treatments significantly decreased the ET levels in rats fed HFD, and regarding this effect, LGA was even superior to simvastin. The lower serum ET may result in lower levels of inflammatory factors. SCFAs are involved in maintaining intestinal homeostasis and controlling gut leakage. We found that butyric acid was the highest among the SCFAs and it was significantly elevated by LGA administration. Zhuang Li et al. reported that butyrate administration prevented obesity, hyperinsulinemia, hypertriglyceridemia, and hepatic steatosis in mouse models [20]. The improvement of hyperlipidemia and fatty liver caused by LGA may be partially attributed to the elevated butyric acid.

Bile acid metabolism plays a key role in the metabolism of cholesterol. Blood cholesterol is transported into the liver, where cholesterol is synthesized into CA and CDCA (also including αMCA and βMCA in mice) via the classical or alternative pathway. The free bile acids are conjugated with taurine or glycine to form conjugated bile acids including TCA/GCA and TCDCA/GCDCA (also including TαMCA/GαMCA in mice). The primary bile acids are released into the duodenum after the ingestion of a meal. Some conjugated bile acids can be hydrolyzed into free bile acids by BSH produced by BSH-producing microbiota in the intestine. Some primary bile acids can be transformed into secondary bile acids via dehydrogenation or dihydroxylation by gut microbiota. A total of 95% of bile acids in the ileum can be transported back to the liver through a process termed enterohepatic circulation. The remaining 5% of bile acids are excreted in the feces [21]. At the same time, approximately 5% of bile acids are synthesized from cholesterol in the liver to make up for the loss of bile acids. The more bile acids were excreted in feces, the more bile acids were synthesized in the liver. In our study, the levels of total bile acids in the feces of LGA group were significantly increased compared with the PBS group. The cholesterol-lowering property of LGA may in part be due to the increased excretion of bile acids in feces. Interestingly, the conjugated bile acids in the feces of LGA rats were significantly reduced compared with the PBS group, which may be attributed to the enhanced BSH activity caused by LGA administration. Although LGA failed to colonize the intestine, the daily administration of LGA may increase the BSH activity in the gut and facilitate the hydrolysis of conjugated bile acids into free bile acids. Consistent with the profiles of bile acids in feces, the free bile acids in serum of LGA group, such as 23-norcholic acid, 7-ketolithocholic acid, beta-muricholic acid, cholic acid, and deoxycholic acid, were increased, while conjugated bile acids, such as glycoursodeoxycholic acid and tauro-alpha-muricholic acid sodium salt, were decreased. It has been reported that a strain of *Lactobacillus casei* has a cholesterol-lowering characteristic, mainly depending on the BSH in a hamster model [22] and fucoidan and galactooligosaccharides ameliorates dyslipidemia in rats through enhancing the BSH activity of gut microbiota [23]. However, another study reported that theabrownin attenuated hypercholesterolemia through reducing BSH-enriched bacteria and suppressing BSH activity in a C57BL/6J mouse model [24]. The discrepant results in different studies may be due to the use of different animal models. A double-blind, placebo-controlled, randomized, parallel-arm, and multi-center study showed that a BSH-producing *Lactobacillus reuteri* NCIMB 30,242 was effective and safe for lowering cholesterol in hypercholesterolaemic subjects [25]. Numerous clinical studies have demonstrated that BSH-active probiotics are efficient in lowering total cholesterol [26]. However, studies also suggest that BSH may act as a double sword, as Jones ML reported that BSHs could have both beneficial and harmful effects on host health: on the one hand, BSH activity reduces serum cholesterol levels, while on the other hand, it also causes lipid metabolism disorders [26]. Further studies are necessary to explore the underlying mechanisms concerning how BSH plays its role in hyperlipidemia.

We also found that LGA administration improved the ratio of secondary/primary bile acids, which helps maintain the homeostasis of gut microbiota. In obesity patients, the abundance of phylum of Firmutes and the ratio of Firmutes/Bacteroids were decreased compared to healthy individuals [4], and similar results were seen in rats fed HFD, indicating the dysbiosis existing in obesity and hyperlipidemia patients. In the current study, LGA intervention decreased the abundance of phylum of Firmutes and the ratio of Firmutes/Bacteroids caused by the HFD. It was reported that *Bacteroides thetaiotaomicron* was markedly reduced in obese individuals, and gavage with *B. thetaiotaomicron* reduced body-weight gain and adiposity in mice [27]. *Parabacteroides distasonis* alleviated obesity and metabolic dysfunctions in HFD-fed mice [28]. We found that LGA administration significantly elevated the relative abundance of the genera of Bacteroides and Parabacteroides, which may contribute to its effect on improving hyperlipidemia. Of note, the relative abundance of *Blautia* genus increased nearly 10-fold after LGA administration compared to PBS control. *Blautia producta* has been shown to alleviate the progress of the non-alcoholic liver disease (NAFLD) [29]. Since *Blautia* possesses a series of potential probiotic properties involved in the improvement of metabolic and inflammatory diseases, it is proposed to be a potential probiotic [30]. Thus, the observed alleviation in fatty liver and hyperlipidemia in HCD rats treated with LGA may be, in part, attributed to the increased *Blautia*.

In conclusion, the current study demonstrated that LGA administration improved the blood lipids and fatty liver, which may be attributed to BSH that are known to hydrolyze conjugated bile acids to generate free bile acids, and thus, more bile acids, especially the free bile acids were excreted. Further, the LGA-driven alteration in gut microbiota, such as Bacteroides, Parabacteroides, and *Blautia*, may also play a role in alleviating hyperlipidemia. While the cholesterol-lowering effect of LGA reported in this study is based on the diet-induced hyperlipidemia animal model, the positive findings shed light on its potential application in humans, which thus warrants future clinical trials for verification.

## Figures and Tables

**Figure 1 nutrients-14-04945-f001:**
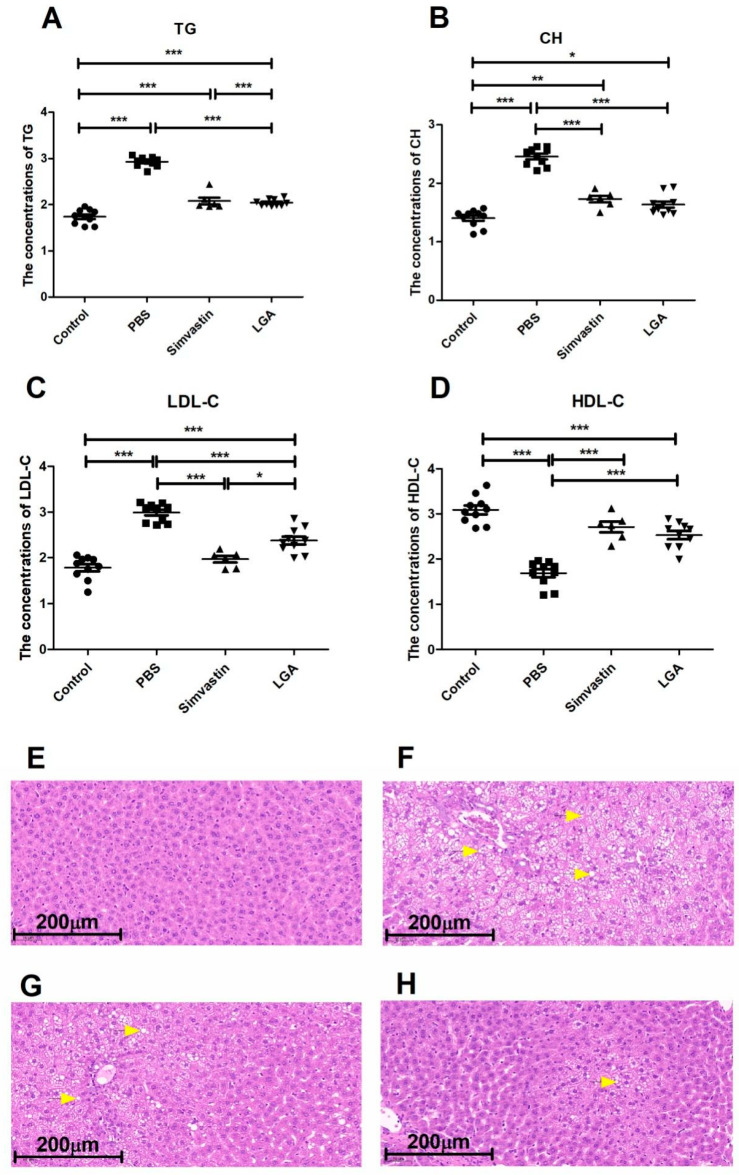
*L. gasseri* administration improved the hyperlipidemia in rat model. *L. gasseri* decreased the concentrations of TG (**A**), CH (**B**), LDL-C (**C**), and increased the level of HDL-C (**D**). (**E**–**H**) were representative pictures of liver HE staining of Control, PBS, Simvastin, and LGA, respectively. The circular vacuoles indicated by the yellow arrow were fat globules. The data were analyzed by the analysis of variance (ANOVA). *p* < 0.05 means the significantly difference. * *p* < 0.05, ** *p* < 0.01, *** *p* < 0.001.

**Figure 2 nutrients-14-04945-f002:**
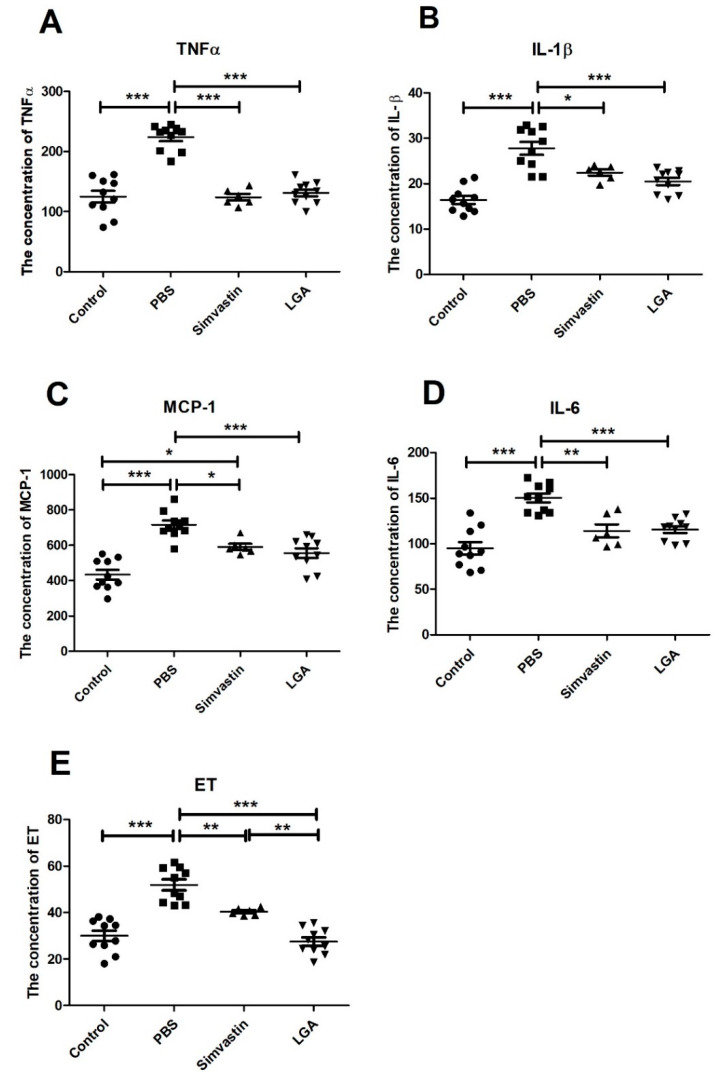
The effect of *L. gasseri* on inflammatory factors in the serum of hyperlipidemia rat. (**A**–**E**) represents the levels of TNFα, IL-1β, MCP-1, IL-6, and ET. The data were analyzed by analysis of variance (ANOVA). *p* < 0.05 means the significant difference. * *p* < 0.05, ** *p* < 0.01, *** *p* < 0.001.

**Figure 3 nutrients-14-04945-f003:**
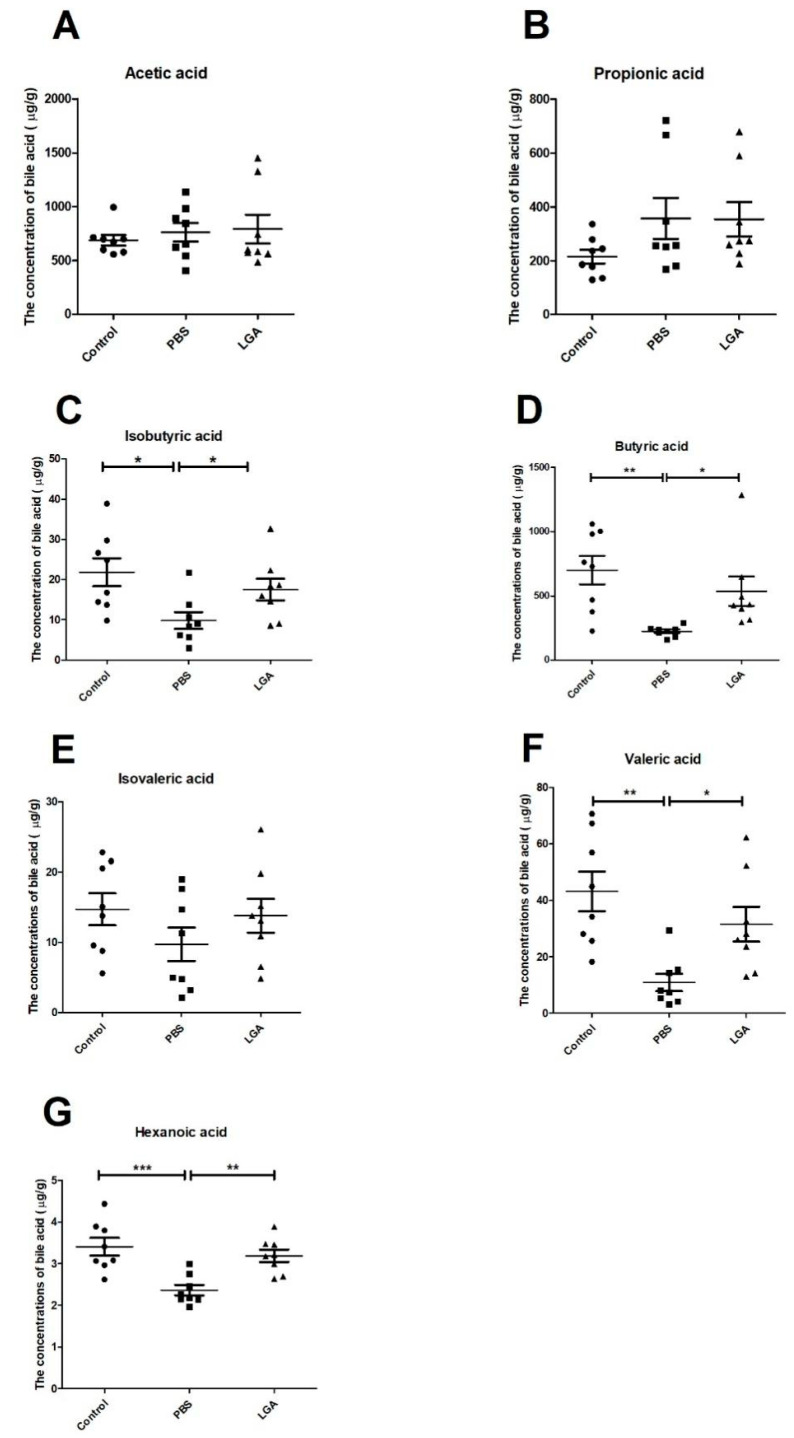
The effect of *L. gasseri* on the short-chain fatty acids (SCFAs) in the feces of hyperlipidemia rats. (**A**–**G**) represent the concentrations of acetic acid, propionic acid, isobutyric acid, butyric acid, isovaleric acid, valeric acid, and hexanoic acid, respectively. The data were analyzed by analysis of variance (ANOVA). * *p* < 0.05, ** *p* < 0.01, *** *p* < 0.001.

**Figure 4 nutrients-14-04945-f004:**
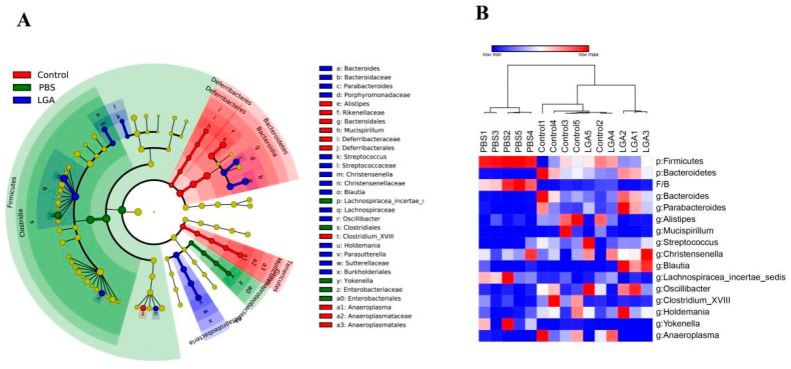
LEFSE and heatmap of differential OTUs. (**A**) LEFSE. (**B**) Heatmap of differential OTUs.

**Figure 5 nutrients-14-04945-f005:**
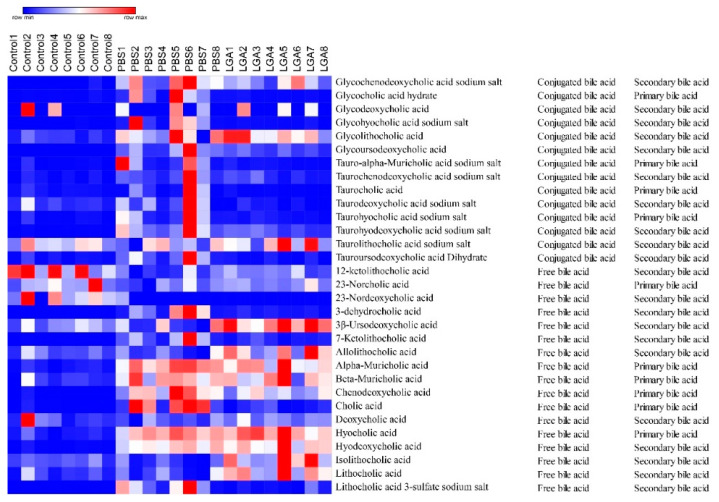
The heatmap of bile acids in the feces of hyperlipidemia rat.

**Figure 6 nutrients-14-04945-f006:**
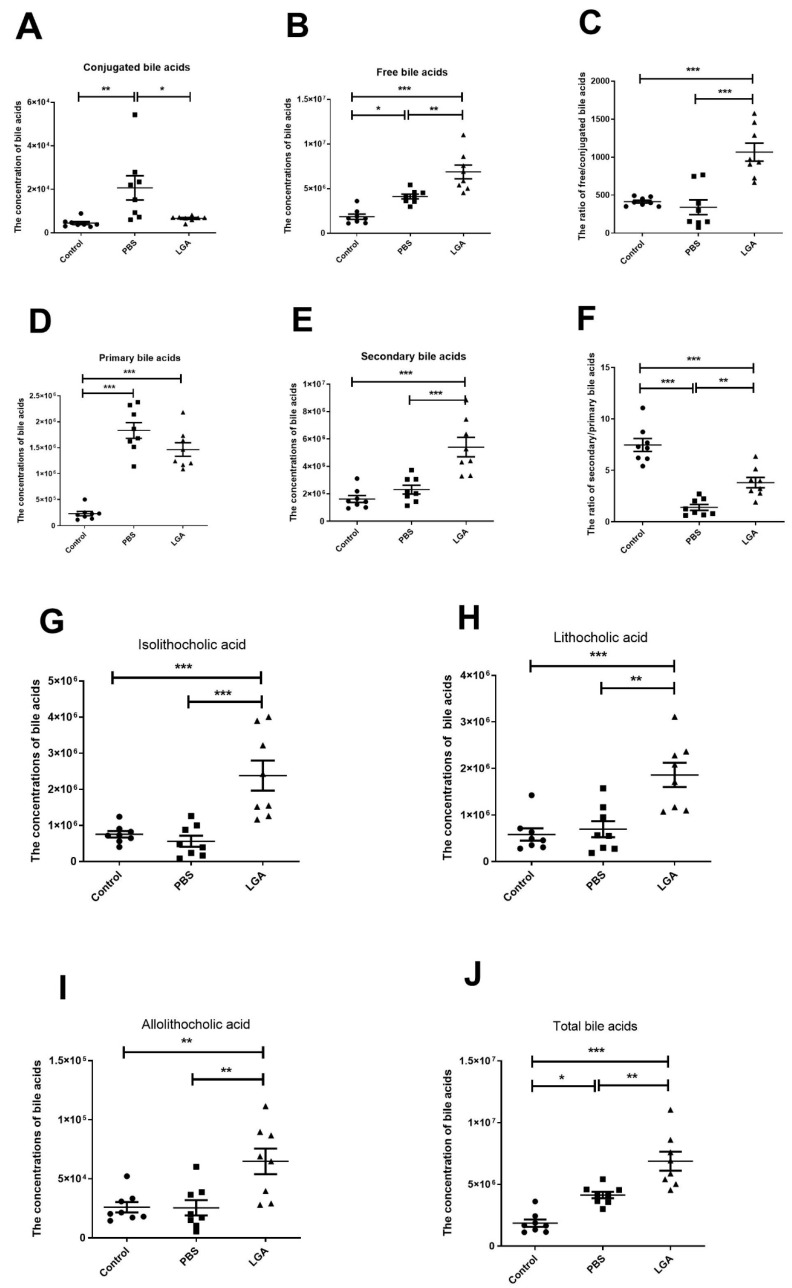
The effect of *L. gasseri* on conjugated/free bile acids and primary/secondary bile acids in the feces of rat. (**A**–**C**) are conjugated bile acid, free bile acid, and the ratio of free/conjugated bile acids, respectively. (**D**–**F**) are primary bile acids, secondary bile acids, and the ratio of secondary/primary bile acids, respectively. (**G**–**J**) represent the concentrations of lithocholic acid, isolithocholic acid, and allolithochoic acid, respectively. The data were analyzed by analysis of variance (ANOVA). * *p* < 0.05, ** *p* < 0.01, *** *p* < 0.001.

**Figure 7 nutrients-14-04945-f007:**
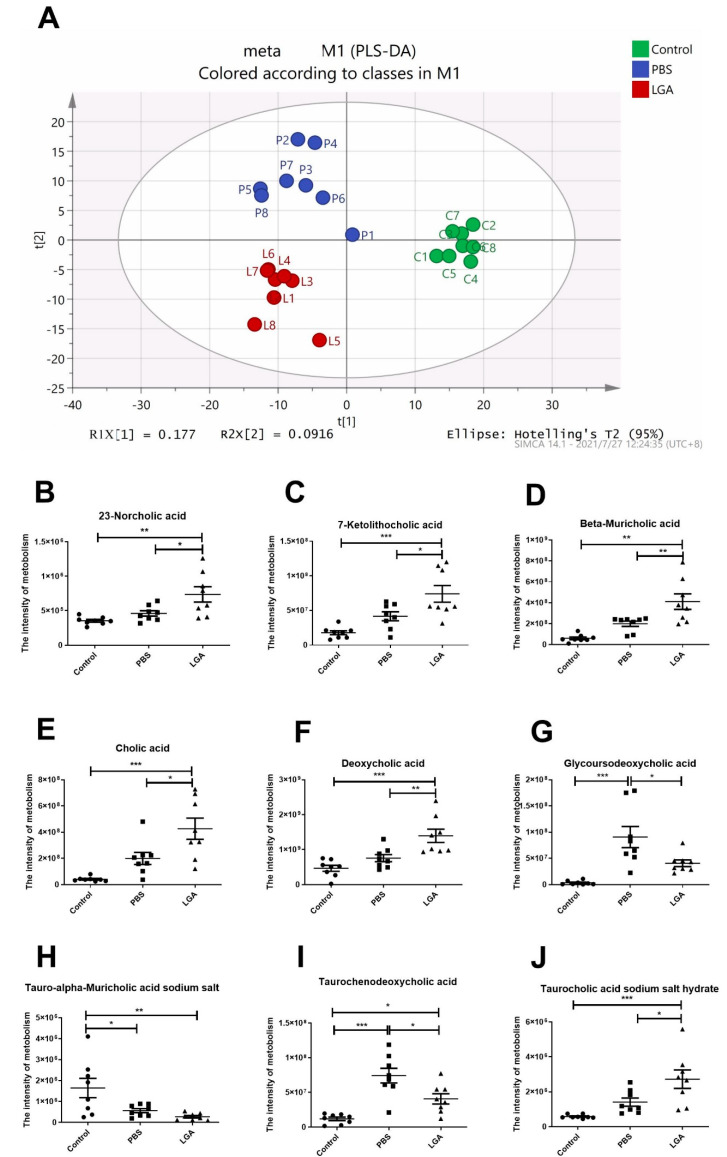
The Effect of L. gasseri RW2014 on Serum Metabolism in Rat. (**A**) PLS-DA score plot of serum metabolite profile in three independent groups. (**B**–**J**) represent the intensities of 23−norcholic acid, 7−ketolithocholic acid, beta−muricholic acid, cholic acid, deoxycholic acid, glycoursodeoxycholic acid, tauro−alpha−muricholic acid sodium salt, taurochenodeoxycholic acid, and taurocholic acid sodium salt hydrate, respectively. The data were analyzed by an analysis of variance (ANOVA). * *p* < 0.05, ** *p* < 0.01, *** *p* < 0.001.

## Data Availability

All data included in this study are available upon request by contact with the corresponding author.

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
