# Peer review of "Lactobacillus gasseri RW2014 Ameliorates Hyperlipidemia by Modulating Bile Acid Metabolism and Gut Microbiota Composition in Rats"

_nutrients, 2022, doi:10.3390/nu14234945_

Round 1
Reviewer 1 Report
The text of the report looks fine. I have noticed a few editorial mistakes. However, THERE NO RESULTS included. The submited item does not FIGURES and I cannot estimate the study sufficiently !!!
So, I am waiting for complete version of the study to be be reviewed.
Author Response
We apologized for not including the figures in the text of manuscript and did not find where to submit the figures. We have revised the manuscript using the template as recommended which included figures.
Reviewer 2 Report
In the present study, authors attempted to use a rat model with hyperlipidemia to evaluate the efficacy of Lactobacillus gasseri RW2014 in ameliorating fatty liver and altering the gut microbiota. Although the results of this study may provide meaningful information to readers, poor manuscript preparation limits its importance.
1. All figures are missing in the present manuscript.
2. There are several typos in the manuscript, such as “Aling” and “stud” in line 65.
Author Response

(The authors gave the same response as above.)

Round 2
Reviewer 1 Report
Yes, now having reciving the previously lacking information I see the results supporting each other and final conclusions. Hence I recommend to accept the study to publication.
Author Response
Dear editors:
We appreciated reviewer’s comment to our manuscript entitled “Lactobacillus gasseri RW2014 ameliorates hyperlipidemia by modulating bile acid metabolism and gut microbiota composition in rats”.
In order to polish the language of this manuscript, we had our manuscript revised by an English-speaking professor. All the revisions were marked by using the “Track Changes” function in the re-submitted manuscript.
Sincerely